# Fast Attention Requires Bounded Entries

Josh Alman[*]                           Zhao Song[†]

## Abstract

In modern machine learning, inner product attention computation is a fundamental task for training large language models such as Transformer, GPT-1, BERT, GPT-2, GPT-3 and ChatGPT. Formally, in this problem, one is given as input three matrices $Q, K, V \in [-B, B]^{n \times d}$, and the goal is to construct the matrix $\mathrm{Att}(Q, K, V) := \mathrm{diag}(A\mathbf{1}_n)^{-1} AV \in \mathbb{R}^{n \times d}$, where $A = \exp(QK^\top/d)$ is the 'attention matrix', and $\exp$ is applied entry-wise. Straightforward methods for this problem explicitly compute the $n \times n$ attention matrix $A$, and hence require time $\Omega(n^2)$ even when $d = n^{o(1)}$ is small.

In this paper, we investigate whether faster algorithms are possible by *implicitly* making use of the matrix $A$. We present two results, showing that there is a sharp transition at $B = \Theta(\sqrt{\log n})$.

- If $d = O(\log n)$ and $B = o(\sqrt{\log n})$, there is an $n^{1+o(1)}$ time algorithm to approximate $\mathrm{Att}(Q, K, V)$ up to $1/\mathrm{poly}(n)$ additive error.
- If $d = O(\log n)$ and $B = \Theta(\sqrt{\log n})$, assuming the Strong Exponential Time Hypothesis from fine-grained complexity theory, it is impossible to approximate $\mathrm{Att}(Q, K, V)$ up to $1/\mathrm{poly}(n)$ additive error in truly subquadratic time $n^{2-\Omega(1)}$.

This gives a theoretical explanation for the phenomenon observed in practice that attention computation is much more efficient when the input matrices have smaller entries.

## 1 Introduction

Large language models (LLMs) such as Transformer [VSP+17], BERT [DCLT18], GPT-3 [BMR+20], PaLM [CND+22], and OPT [ZRG+22] can process natural language more effectively than smaller models or traditional algorithms. This means that they can understand and generate more complex and nuanced language, which can be useful for a variety of tasks such as language translation, question answering, and sentiment analysis. LLMs can also be adapted to multiple purposes without needing to be retained from scratch. Their power is particularly exemplified by the recent success of ChatGPT, a chat software by OpenAI built on top of GPT-3 [Ope22].

The key technical backbone of LLMs is the *attention matrix* [VSP+17, RNS+18, DCLT18, RWC+19, BMR+20]. An attention matrix is a square matrix whose rows and columns correspond to words or "tokens", and whose entries correspond to the correlations between these tokens in natural text. The attention matrix is then used to calculate the importance of each input token in a sequence when producing an output. In an attention mechanism, each input token is given a weight or score, which reflects its importance or relevance to the current output being generated. These scores are calculated based on a comparison between the current output state and the input states, using a similarity function.

---

[*] josh@cs.columbia.edu. Columbia University.

[†] zsong@adobe.com. Adobe Research.

37th Conference on Neural Information Processing Systems (NeurIPS 2023).

More formally, the attention matrix is defined as follows. Let $Q \in \mathbb{R}^{n \times d}$ be the matrix of query tokens, and $K \in \mathbb{R}^{n \times d}$ be the matrix of key tokens. (We focus here on the case when $d = n^{o(1)}$, so $d \ll n$.) The attention matrix is an $n \times n$ matrix $A$ where the rows and columns correspond to the input tokens in the sequence. Each entry in the matrix represents the attention weight or score between a particular input token (query token $Q$) and a particular output token (key token $K$). The diagonal entries of the matrix represent self-attention scores, which measure the importance of each token with respect to itself.

The major bottleneck to speeding up LLM operations (in the case of modeling long sequences with large $n$) is the time to perform attention matrix computations [VSP+17, RNS+18, DCLT18, RWC+19, BMR+20, WLK+20, KKL20]. These computations ask us to multiply the attention matrix $A$ with another value token matrix $V \in \mathbb{R}^{n \times d}$.

We formally define Attention computation as follows. Throughout this paper, we write $\exp$ to denote the *entry-wise* exponential for matrices.

**Definition 1.1** (Exact Attention Computation $\mathsf{EAttC}(n, d)$). *Given three matrices $Q, K, V \in \mathbb{R}^{n \times d}$, output the $n \times d$ matrix $\mathrm{Att}(Q, K, V)$ defined by*

$$\mathrm{Att}(Q, K, V) := D^{-1} A V$$

*where $A \in \mathbb{R}^{n \times n}$ and diagonal matrix $D \in \mathbb{R}^{n \times n}$ are defined as*

$$A := \exp(QK^\top / d), \quad and \quad D := \mathrm{diag}(A \mathbf{1}_n).$$

The straightforward algorithm for this problem computes the matrix $A$ and then performs the multiplications $D^{-1} A V$, in time $n^{2+o(1)}$. Since $A$ is an $n \times n$ matrix with $n^2$ entries, it is impossible to improve on this much while explicitly computing the matrix $A$. However, the input to the problem is not $A$, but rather the three matrices $Q, K, V$ which each have only $n^{1+o(1)}$ entries. An algorithm which only *implicitly* makes use of $A$, without explicitly computing all its entries, could hope to run in almost linear time!

In this paper, we investigate the possibility of accelerating attention computations in this way. The two main questions we address are:

- **Q1.** When can we perform attention computations in almost linear time $n^{1+o(1)}$?
- **Q2.** When can we prove that subquadratic-time algorithms for attention computations are *impossible*?

In most LLMs, it suffices to *approximately* perform attention computations throughout the inference process as long as there are reasonable precision guarantees [CGRS19, KKL20, WLK+20, DKOD20, KVPF20, CDW+21, CDL+22, LWD+23, ZSZ+23]. We therefore focus here on approximate attention computation, which can potentially be performed even faster than exact computation. Mathematically, we define the *approximate* version of $\mathsf{EAttC}$ as follows.

**Definition 1.2** (Approximate Attention Computation $\mathsf{AAttC}(n, d, B, \epsilon_a)$). *Let $\epsilon_a > 0$ and $B > 0$ be parameters. Given three matrices $Q, K, V \in \mathbb{R}^{n \times d}$, with the guarantees that $\|Q\|_\infty \leq B$, $\|K\|_\infty \leq B$, and $\|V\|_\infty \leq B$, output a matrix $T \in \mathbb{R}^{n \times d}$ which is approximately equal to $D^{-1} A V$, meaning,*

$$\|T - D^{-1} A V\|_\infty \leq \epsilon_a.$$

*Here, for a matrix $M \in \mathbb{R}^{n \times n}$, we write $\|M\|_\infty := \max_{i,j} |M_{i,j}|$.*

Again, the straightforward algorithm for this problem runs in time $O(n^2 d) \leq n^{2+o(1)}$, but the input size is only $O(nd) \leq n^{1+o(1)}$. Our goal is to investigate when faster algorithms are possible in terms of the parameters $d$, $B$, and $\epsilon_a$.

## 1.1 Our Results

We focus on the natural setting where $d = O(\log n)$ (the setting where we model long sequences) and $\epsilon_a = 1/\mathrm{poly}(n)$ (low enough error so that attention computations over an entire network can be combined). Our main results show that whether or not there is a fast algorithm for $\mathsf{AAttC}$ critically depends on $B$, the magnitudes of the entries in the input matrices.

We first show a lower bound, that when $B \geq \Omega(\sqrt{\log n})$, it is impossible to design a truly subquadratic-time algorithm. Our lower bound makes use of the Strong Exponential Time Hypothesis (SETH) [IP01], a popular conjecture [Wil18] from the area of fine-grained complexity regarding the time required to solve $k$-SAT. (See Section 4 below where we discuss SETH in more detail.)

**Theorem 1.3** (Lower bound, informal version of Theorem 4.6). *Assuming* SETH*, for every $q > 0$, there are constants $C, C_a, C_b > 0$ such that: there is no $O(n^{2-q})$ time algorithm for the problem* AAttC$(n, d = C \log n, B = C_b \sqrt{\log n}, \epsilon_a = n^{-C_a})$.

Our second complementary result is a new algorithm, showing that when $B < o(\sqrt{\log n})$, the problem can be solved very efficiently, in almost linear time.

**Theorem 1.4** (Upper bound, informal version of Theorem 3.8). *There is an algorithm (Algorithm 1) that solves* AAttC$(n, d = O(\log n), B = o(\sqrt{\log n}), \epsilon_a = 1/\operatorname{poly}(n))$ *in time $n^{1+o(1)}$*.

Our Theorems 1.3 and 1.4 show that the attention computation problem AAttC exhibits a very tight transition at $B = \Theta(\sqrt{\log n})$ from almost linear time to trivial quadratic time. When $B < o(\sqrt{\log n})$ is smaller, the problem can be solved in almost linear time $n^{1+o(1)}$ in the input size, using our algorithm for Theorem 1.4. When $B \geq \Omega(\sqrt{\log n})$ is greater, our algorithm from Theorem 1.4 no longer applies, and furthermore our lower bound from Theorem 1.3 shows that it is *impossible* to solve the problem in truly subquadratic time, no matter what algorithmic techniques one uses (assuming SETH).

It has been observed in LLM implementations in practice that computations are much faster when one assumes that the matrix entries are bounded or can be well-approximated using a small number of bits (see, e.g., [ZBIW19, Section 2] and [KVPF20, Section 3.2.1]). Our work can be viewed as giving a theoretical explanation for this phenomenon, and helping to explain why techniques like quantization [ZBIW19] and low-degree polynomial approximation [KVPF20] have been so effective in practice.

**Related Work.**

A recent work by Zandieh, Han, Daliri, and Karbasi [ZHDK23] was the first to give an algorithm with provable guarantees for attention approximation. Their algorithm makes use of locality sensitive hashing (LSH) techniques [CKNS20] which, as we will discuss next, is quite different from our algorithm for Theorem 1.4 which uses the polynomial method [ACSS20, AA22].

In the case when $d = o(\log^2 n)$, they achieve a running time of roughly $O(n^{1.17} \cdot d/\epsilon_r^2)$, where $\epsilon_r$ is a *relative* error parameter (which is similar, though not exactly the same, as our $\epsilon_a$ from Definition 1.2). In particular, their algorithm applies for larger $d$ than ours (we require $d = O(\log n)$), but we achieve almost linear time $n^{1+o(1)}$ (whereas their running time is bounded below by $\Omega(n^{1.17})$), and our algorithm can handle any polynomial error $\epsilon_a = 1/\operatorname{poly}(n)$ (whereas they require $\epsilon_r \geq 1/n^{o(1)}$ to not increase the running time by a polynomial factor).

It is natural to wonder whether further improvements are possible by combining our techniques with those of [ZHDK23]. However, our lower bound of Theorem 1.3 shows that our algorithm of Theorem 1.4 is already essentially tight and cannot be substantially improved.

Another recent work by Keles, Wijewardena, and Hedge [KWH23] was the first to prove a lower bound for attention computation assuming SETH. They prove, among other results, that AAttC cannot be solved in truly subquadratic time in the case when $d = \omega(\log n)$. Our Theorem 1.3 improves their result to also hold for $d = \Theta(\log n)$, and to show how the complexity changes with the magnitude of entries $B$ (which is not studied by [KWH23]). As we discuss more shortly, both our lower bound proof and [KWH23] use the high-level technique of [BIS17], although our more fine-grained analysis of the parameters $d, B$ requires a more intricate analysis and the use of other techniques from fine-grained complexity related to approximate nearest neighbor search [Rub18] and the polynomial method [AA22].

## 1.2 Technique Overview

Our high-level approach is to make use of similarities between attention computation and other computational problems related to Kernel Density Estimation (KDE). Such a relationship was

investigated by recent work [TBY$^+$19, ZHDK23]. In particular, [ZHDK23] was inspired to apply LSH techniques to attention computation because of the prevalence of LSH in KDE algorithms [CS17, BCIS18, CS19, CKNS20]. The main conceptual idea behind our results is that different techniques from the KDE literature, other than LSH, can be modified to apply in this setting and yield tight algoriths and lower bounds.

To design our algorithm for Theorem 1.3, we instead build off of a different line of work on KDE which makes use of the 'polynomial method in algorithm design'. Suppose $M \in \mathbb{R}^{n \times n}$ is a matrix, $f : \mathbb{R} \to \mathbb{R}$ is a function, and let $f(M)$ denote the matrix one gets by applying $f$ entry-wise to $M$. The polynomial method is a technique for finding low-rank approximations of $f(M)$. It shows that if $M$ has low rank, and if $f$ can be approximated by a low-degree polynomial, then the matrix $f(M)$ is very close to a low-rank matrix whose low-rank decomposition can be computed efficiently.

To use this to solve AAttC, we make use of a recent result which bounds the degree required to approximate the exponential function by a polynomial [AA22] in order to find a low-rank approximation of the attention matrix $A$. Prior work [ACSS20, ACM$^+$20, AA22] applied these polynomials in a similar way to solve the Gaussian KDE problem; our main observation is that by an appropriate rescaling, this approach can be modified to apply to AAttC as well.

The proof of our lower bound Theorem 1.3 builds off of another line of work on the fine-grained complexity of KDE problems [BIS17, ACSS20, AA22]. The main idea is to give a fine-grained reduction from the well-studied problem of Approximate Nearest Neighbor search ANN. In ANN, one is given as input $n$ vectors of dimension $d$, and an error parameter $\epsilon > 0$, and the goal is to find a pair of vectors whose distance is at most $(1 + \epsilon)$ times the *minimum* distance between any pair of the vectors. The straightforward algorithm for ANN runs in quadratic time, and it is known that it is impossible to solve ANN in truly subquadratic time assuming SETH [Rub18].

In order to prove our lower bound, we show that AAttC can be used to solve ANN. The key idea is that, if the matrices $Q$ and $K$ from AAttC are formed by concatenating the input vectors to the ANN problem, then the nearest neighbor vectors correspond to the largest entries of the attention matrix $A$. It is not immediately clear that AAttC can be used to detect large entries of $A$, since the output is rescaled by the matrix $D^{-1}$, but we show that this can be overcome with some modifications to the input vectors which approximately balance the rows of $A$. Prior work [BIS17, ACSS20, AA22] used a very similar approach to give lower bounds for KDE problems, although KDE doesn't involve any rescaling factors.

**Roadmap.**

In Section 2, we introduce relevant notation and tools from prior work. In Section 3, we present and analyze our attention algorithm. In Section 4, we prove our fine-grained attention lower bound. In Section 5, we provide a conclusion for this paper.

## 2   Preliminaries

We work in the standard real-RAM model and assume arithmetic operations on real numbers can be performed in constant time in our algorithms.

We use $\mathcal{T}_{\mathrm{mat}}(a, b, c)$ to denote the time to multiply an $a \times b$ matrix with another $b \times c$ matrix. In fact, we will only make use of the straightforward, practical bound $\mathcal{T}_{\mathrm{mat}}(a, b, c) \leq O(abc)$. In principle, fast theoretical matrix multiplication algorithms could be used instead to improve this bound and speed up our algorithms here (in exchange for making them less practical). That said, because of our parameter settings[3], we will see that faster matrix multiplication could only improve low-order terms in our running times.

For any positive integer, we use $[n]$ to denote set $\{1, 2, \cdots, n\}$.

For a matrix $M$, we write $\|M\|_\infty$ to denote its $\ell_\infty$ norm, i.e., $\|M\|_\infty := \max_{i,j} |M_{i,j}|$. For a matrix $M$, we use $M^\top$ to denote its transpose.

We use $\mathbf{1}_n$ to denote a length-$n$ vector whose entries are all 1s. We use $\mathbf{0}_n$ to denote a length-$n$ vector whose entries are all 0s.

---

[3]We will make use of $\mathcal{T}_{\mathrm{mat}}(n, n^{o(1)}, n^{o(1)})$, which can be solved straightforwardly in time $n^{1+o(1)}$, and which cannot be solved much faster since it has input size $n^{1+o(1)}$.

For any matrix $A \in \mathbb{R}^{n \times n}$, we use $\exp(A) \in \mathbb{R}^{n \times n}$ to denote the matrix where $\exp(A)_{i,j} = \exp(A_{i,j})$. In other words, all the $\exp()$ operators in this paper are applied entry-wise to matrices. In particular, we will not use matrix exponentials in this paper.

For a vector $x \in \mathbb{R}^n$, we use $\|x\|_0$ to denote its number of non-zero entries, we use $\|x\|_1$ to denote its $\ell_1$ norm, i.e., $\|x\|_1 := \sum_{i=1}^n |x_i|$, and we use $\|x\|_2$ to denote its $\ell_2$ norm, i.e., $\|x\|_2 := (\sum_{i=1}^n |x_i|^2)^{1/2}$. For a vector $x$, we use $x^\top$ to denote its transpose.

## 2.1 Additive Error for Polynomial Approximation

Our algorithm for attention computation will critically make use of a polynomial approximation for the exponential function. In particular, we use the following tight construction from previous work [AA22].

**Lemma 2.1** ([AA22]). *Let $B > 1$ and let $\epsilon \in (0, 0.1)$. There is a polynomial $P : \mathbb{R} \to \mathbb{R}$ of degree $g := \Theta\left(\max\left\{\frac{\log(1/\epsilon)}{\log(\log(1/\epsilon)/B)}, B\right\}\right)$ such that for all $x \in [0, B]$, we have*

$$|P(x) - \exp(x)| < \epsilon.$$

*Moreover, $P$ can be computed efficiently: its coefficients are rational numbers with $\mathrm{poly}(g)$-bit integer numerators and denominators which can be computed in $\mathrm{poly}(g)$ time.*

## 2.2 From Additive Error to Relative Error

We note that in our setting, Lemma 2.1 can be used to give a relative error approximation as well:

**Corollary 2.2.** *Let $B > 1$ and let $\epsilon \in (0, 0.1)$. There is a polynomial $P : \mathbb{R} \to \mathbb{R}$ of degree $g := \Theta(\max\{\frac{\log(1/\epsilon)}{\log(\log(1/\epsilon)/B)}, B\})$ such that for all $x \in [-B, B]$, we have*

$$|P(x) - \exp(x)| < \epsilon \cdot \exp(x).$$

*Proof.* By Lemma 2.1, there is a polynomial $Q : \mathbb{R} \to \mathbb{R}$ of degree $g = \Theta(\{\frac{\log(1/\epsilon)}{\log(\log(1/\epsilon)/B)}, B\})$ such that, for all $y \in [0, 2B]$ we have $|Q(y) - \exp(y)| \leq \epsilon$. Our desired polynomial is the rescaled $P(x) := Q(x + B)/\exp(B)$. Indeed, for any $x \in [-B, B]$, we have $\exp(x) \geq \exp(-B)$, and so

$$
\begin{aligned}
|P(x) - \exp(x)| &= |Q(x + B)/\exp(B) - \exp(x)| \\
&= |Q(x + B) - \exp(x + B)|/\exp(B) \\
&\leq \epsilon/\exp(B) \\
&\leq \epsilon \cdot \exp(x),
\end{aligned}
$$

as desired. $\square$

# 3 Attention Algorithm

In this section, we show how to use polynomial approximations for the exponential function in order to approximately perform attention computations. In Section 3.1, we define the type of low-rank matrix approximation which we will use. In Section 3.2, we show how polynomial approximations can give rise to such low-rank matrix approximations. In Section 3.3, we bound the entries of the matrix $QK^\top \in \mathbb{R}^{n \times n}$ (before converting it to the attention matrix) to confirm that our polynomial approximation applies. In Section 3.4, we state our main technique for approximating the attention matrix. In Section 3.5, we show how to control the error propagation from $A$ to the rescaling matrix $D$. In Section 3.6, we further explain how to control the error propagation from $D$ and $A$ to the resulting attention matrix. Finally, in Section 3.7, we conclude our general algorithm, and in Section 3.8, we appropriately select the parameters to achieve almost linear time.

## 3.1 Matrix Low-Rank Approximation

**Definition 3.1.** *Let $r \geq 1$ denote a positive integer. Let $\epsilon \in (0, 0.1)$ denote an accuracy parameter. Given a matrix $A \in \mathbb{R}_{\geq 0}^{n \times n}$, we say $\widetilde{A} \in \mathbb{R}_{\geq 0}^{n \times n}$ is an $(\epsilon, r)$-approximation of $A$ if*

- $\widetilde{A} = U_1 \cdot U_2^\top$ *for some matrices* $U_1, U_2 \in \mathbb{R}^{n \times r}$ *(i.e.,* $\widetilde{A}$ *has rank at most* $r$*), and*

- $|\widetilde{A}_{i,j} - A_{i,j}| \le \epsilon \cdot A_{i,j}$ *for all* $(i,j) \in [n]^2$.

## 3.2 From Low Degree Polynomials to Low Rank Matrices

**Lemma 3.2.** *Let* $M = XY^\top \in \mathbb{R}^{n \times n}$ *denote a matrix with* $X, Y \in \mathbb{R}^{n \times d}$*. Let* $P(x)$ *denote a degree-$g$ polynomial, and define* $r = \binom{2(g+d)}{2g}$.

*There is an algorithm that runs in* $O(nrg)$ *time and, given as input the matrix* $X, Y$*, constructs matrices* $U_1, U_2 \in \mathbb{R}^{n \times r}$ *such that* $P(M) = U_1 U_2^\top$*. (Here,* $P(M)$ *denotes the entry-wise application of* $P$ *to* $M$*.)*

Due to space limitation, we defer the proof of Lemma 3.2 to Appendix A.

## 3.3 Matrix $QK^\top$ Has Bounded Entries

**Lemma 3.3** (Bounded entry). *Suppose* $B \ge 1$ *and matrices* $Q, K \in \mathbb{R}^{n \times d}$ *have* $\|Q\|_\infty \le B$ *and* $\|K\|_\infty \le B$*. Then, we have*

$$\|QK^\top/d\|_\infty \le B^2.$$

*Proof.* For each $(i,j) \in [n] \times [n]$, we have

$$
\begin{aligned}
|(QK^\top)_{i,j}| = |\sum_{l=1}^{d} Q_{i,l} K_{j,l}| \\
\le d \cdot \|Q\|_\infty \cdot \|K\|_\infty \\
\le d \cdot B^2,
\end{aligned}
$$

as desired. $\qquad\square$

## 3.4 Key Lemma

Our key lemma shows that, even though the attention matrix $A$ may have full rank, it has a low-rank approximation that is easy to compute:

**Lemma 3.4.** *Suppose* $Q, K \in \mathbb{R}^{n \times d}$*, with* $\|Q\|_\infty \le B$*, and* $\|K\|_\infty \le B$*. Let* $A := \exp(QK^\top/d) \in \mathbb{R}^{n \times n}$*. For accuracy parameter* $\epsilon \in (0,1)$*, there is a positive integer $g$ bounded above by*

$$g = O\Big(\max\Big\{\frac{\log(1/\epsilon)}{\log(\log(1/\epsilon)/B^2)}, B^2\Big\}\Big),$$

*and a positive integer $r$ bounded above by*

$$r \le \binom{2(g+d)}{2g}$$

*such that: There is a matrix* $\widetilde{A} \in \mathbb{R}^{n \times n}$ *that is an* $(\epsilon, r)$*-approximation (Definition 3.1) of* $A \in \mathbb{R}^{n \times n}$*. Furthermore, the matrices* $U_1$ *and* $U_2$ *defining* $\widetilde{A}$ *can be computed in* $O(n \cdot r)$ *time.*

*Proof.* Let $M := QK^\top/d$. From Lemma 3.3, we know that $\|M\|_\infty \le B^2$. Thus, applying Corollary 2.2 (with bound $B^2$ on its entries), there is a degree-$g$ polynomial $P$ such that the matrix $\widetilde{A} = P(M)$ is an $(\epsilon, r)$-approximation to $A$ (See the definition of $(\epsilon, r)$-approximation in Definition 3.1.) We can then compute $U_1, U_2$ using Lemma 3.2, which gives the bound

$$r \le \binom{2(g+d)}{2g}.$$

This completes the proof. $\qquad\square$

### 3.5 From $A$ to $D$

**Lemma 3.5.** *Let $A \in \mathbb{R}^{n \times n}$ be any matrix whose entires are all positive and $\epsilon_A \in (0, 0.1)$ be any parameter. Let $\widetilde{A} \in \mathbb{R}^{n \times n}$ be any matrix such that, for all $(i, j) \in [n] \times [n]$, we have*

$$|\widetilde{A}_{i,j} - A_{i,j}| \leq \epsilon_A \cdot A_{i,j}.$$

*Define the matrices $D, \widetilde{D} \in \mathbb{R}^{n \times n}$ by $D = \operatorname{diag}(A\mathbf{1}_n)$ and $\widetilde{D} = \operatorname{diag}(\widetilde{A}\mathbf{1}_n)$. Then, for all $i \in [n]$ we have*

$$|\widetilde{D}_{i,i} - D_{i,i}| \leq \epsilon_A \cdot D_{i,i}.$$

Due to space limitation, we defer the proof of Lemma 3.5 into Appendix A.

### 3.6 From $A$ and $D$ to Attention Matrix

**Lemma 3.6.** *Let $\epsilon_A, \epsilon_D \in (0, 0.1)$ and $B > 1$ be parameters, and let $V \in \mathbb{R}^{n \times d}$ denote a matrix with $\|V\|_\infty \leq B$. Let $A \in \mathbb{R}^{n \times n}$ be any matrix whose entires are all positive, and let $\widetilde{A} \in \mathbb{R}^{n \times n}$ be a matrix such that, for all $(i, j) \in [n] \times [n]$ we have*

$$|\widetilde{A}_{i,j} - A_{i,j}| \leq \epsilon_A \cdot A_{i,j}.$$

*Let $D, \widetilde{D} \in \mathbb{R}^{n \times n}$ be any diagonal matrices with positive entries on their diagonals, with the property that, for all $i \in [n]$, we have*

$$|\widetilde{D}_{i,i} - D_{i,i}| \leq \epsilon_D \cdot D_{i,i}.$$

*Then, we have*

$$\|\widetilde{D}^{-1}\widetilde{A}V - D^{-1}AV\|_\infty \leq (\epsilon_A + \epsilon_D) \cdot B.$$

Due to space limitation, we delay the proof of Lemma 3.6 to Appendix A.

### 3.7 Main Upper Bound

**Theorem 3.7.** *For positive integers $n, d$ and real parameters $\epsilon > 0$ and $B > 1$, there are positive integers $g = \Theta(\max\{\frac{\log(1/\epsilon)}{\log(\log(1/\epsilon)/B^2)}, B^2\})$ and $r = \binom{2(g+d)}{2d}$ such that: There is an algorithm (Algorithm 1) that runs in $O(\mathcal{T}_{\mathrm{mat}}(n, r, d) + nrg)$ time to solve $\mathsf{AAttC}(n, d, B, \epsilon)$ (Definition 1.2).*

*Proof.* The running time of each step is shown in Algorithm 1; its running time follows from Lemma 3.4. Its correctness follows from Lemma 3.5 and Lemma 3.6. $\qquad \square$

---

**Algorithm 1** Our Algorithm

---

1: **procedure** POLYATTENTION($Q \in \mathbb{R}^{n \times d}, K \in \mathbb{R}^{n \times d}, V \in \mathbb{R}^{n \times d}, n, d, B, \epsilon$)  ▷ Theorem 1.4
2:                              ▷ $\epsilon$ is the accuracy output
3:                             ▷ $\|Q\|_\infty, \|K\|_\infty, \|V\|_\infty \leq B$
4:    $g \leftarrow O(\max\{\frac{\log(1/\epsilon)}{\log(\log(1/\epsilon)/B^2)}, B^2\})$
5:    $r \leftarrow \binom{2(g+d)}{2d}$
6:    Construct $U_1, U_2 \in \mathbb{R}^{n \times r}$ via Lemma 3.4         ▷ $O(nrg)$ time
7:    $\widetilde{w} \leftarrow U_1 \cdot (U_2^\top \mathbf{1}_n)$                ▷ $O(nr)$ time
8:    $\widetilde{D}^{-1} = \operatorname{diag}(\widetilde{w}^{-1})$                 ▷ $O(n)$ time
9:    Compute $U_2^\top V \in \mathbb{R}^{r \times d}$            ▷ Takes $\mathcal{T}_{\mathrm{mat}}(r, n, d)$ time
10:   Compute $U_1 \cdot (U_2^\top V)$             ▷ $\mathcal{T}_{\mathrm{mat}}(n, r, d)$ time
11:   $T \leftarrow \widetilde{D}^{-1} \cdot (U_1 \cdot (U_2^\top V))$           ▷ $O(nd)$ time
12:   **return** $T$                    ▷ $T \in \mathbb{R}^{n \times d}$
13: **end procedure**

---

## 3.8 Proof of Theorem 1.4

**Theorem 3.8** (Upper bound, formal statement of Theorem 1.4). $\mathsf{AAttC}(n, d = O(\log n), B = o(\sqrt{\log n}), \epsilon_a = 1/\operatorname{poly}(n))$ *can be solved in time* $\mathcal{T}_{\mathrm{mat}}(n, n^{o(1)}, d) = n^{1+o(1)}$.

*Proof.* If we select the parameters

$$B = o(\sqrt{\log n}), \quad \epsilon = 1/\operatorname{poly}(n), \quad d = O(\log n)$$

in Theorem 3.7, then we see that

$$
\begin{aligned}
g &= O(\max\{\frac{\log(1/\epsilon)}{\log(\log(1/\epsilon)/B^2)}, B^2\}) \\
&= O(\max\{\frac{\log(n)}{\log(\log(n)/B^2)}, B^2\}) \\
&= O(\max\{\frac{\log n}{\log \log n}, o(\log n)\}) \\
&= o(\log n),
\end{aligned}
$$

where the second step follows from $\epsilon = 1/\operatorname{poly}(n)$ and the third step follows from $B = o(\sqrt{\log n})$. Since $g = o(\log n)$, let us write $g = (\log n)/f$ for some $f = \omega(1)$. We thus have that

$$r = \binom{2(d+g)}{2g} \leq \left(\frac{e(d+g)}{g}\right)^{2g} \leq 2^{O(g \log((\log n)/g))} \leq 2^{O(\log n \log(f)/f)} < 2^{o(\log n)} < n^{o(1)}.$$

The second step follows from the generic bound $\binom{a}{b} \leq (ea/b)^b$ for $1 \leq b \leq a$, and the third step uses that $d = O(\log n)$.

Since $d, r, g$ are all bounded by $n^{o(1)}$, our final running time is $n^{1+o(1)}$ as desired. $\square$

## 4 Hardness

In this section, we prove our fine-grained lower bound for attention computation. In Section 4.1, we state the Strong Exponential Time Hypothesis (SETH), the main hardness assumption we will use. In Section 4.2, we define the approximate nearest neighbor search problem, and its known hardness assuming SETH. Finally, in Section 4.3, we give a reduction from approximate nearest neighbor search to attention computation, which implies our hardness result.

### 4.1 Fine-Grained Hypotheses

The Strong Exponential Time Hypothesis (SETH) was introduced by Impagliazzo and Paturi [IP01] over 20 years ago. It is a strengthening of the P $\neq$ NP conjecture, which asserts that our current best SAT algorithms are roughly optimal:

**Hypothesis 4.1** (Strong Exponential Time Hypothesis (SETH)). *For every $\epsilon > 0$ there is a positive integer $k \geq 3$ such that $k$-SAT on formulas with $n$ variables cannot be solved in $O(2^{(1-\epsilon)n})$ time, even by a randomized algorithm.*

SETH is a popular conjecture which has been used to prove fine-grained lower bounds for a wide variety algorithmic problems. See, for instance, the survey [Wil18].

### 4.2 Nearest Neighbor Search

We will make use of a known relationship between SETH and approximate nearest neighbor search.

**Definition 4.2** (Approximate Hamming Nearest Neighbor Search (ANN)). *For a parameter $\epsilon > 0$, in the $(1 + \epsilon)$-Approximate Hamming Nearest Neighbor Search problem for $n$ vectors of dimension $d$, we are given as input two sets $A, B \subset \{0, 1\}^d$ with $|A| = |B| = n$, and our goal is to find an $a^* \in A$ and $b^* \in B$ satisfying $\|a^* - b^*\|_0 \leq (1 + \epsilon) \cdot \min_{a \in A, b \in B} \|a - b\|_0$.*

(This is sometimes called the 'bichromatic' ANN problem, and a monochromatic version has also been studied; see, for instance, [SM19].) Rubinstein [Rub18] showed that for certain parameters, it is impossible to substantially improve on the straightforward quadratic-time algorithm for ANN assuming SETH:

**Lemma 4.3** ([Rub18]). *Assuming* SETH, *for every* $q > 0$, *there are* $\epsilon \in (0,1)$ *and* $C > 0$ *such that* $(1 + \epsilon)$-*Approximate Hamming Nearest Neighbor Search in dimension* $d = C \log n$ *requires* $\Omega(n^{2-q})$ *time.*

**Remark 4.4.** *We may assume that 4.3 holds even in the special case where each input vector from* $A$ *and* $B$ *has half its entries equal to* 0 *and half equal to* 1. *Indeed, for any vector* $a \in \{0,1\}^d$, *we can construct a new vector* $\widetilde{a} \in \{0,1\}^{2d}$ *given by* $\widetilde{a} = \begin{bmatrix} a^\top & \overline{a}^\top \end{bmatrix}^\top$. *Here* $\overline{a} \in \{0,1\}^d$ *is the binary complement of vector* $a$, *i.e.,* $\overline{a}_i = 1 - a_i$ *for all* $i \in [d]$. *Thus,* $\|\widetilde{a}\|_0 = d$. *We can similarly construct a new vector* $\widetilde{b} \in \{0,1\}^{2d}$ *for each* $b \in B$. *After this transformation, for any* $a \in A$ *and* $b \in B$, *we have* $\|\widetilde{a} - \widetilde{b}\|_0 = 2 \cdot \|a - b\|_0$, *so it suffices to find an approximate nearest neighbor among these transformed vectors.*

For convenience in our the analysis, we define a gap version of approximate nearest neighbor search problem $\mathsf{Gap-ANN}(n, d, t, \epsilon)$.

**Definition 4.5** (Gap approximate nearest neighbor search ($\mathsf{Gap-ANN}(n, d, t, \epsilon)$)). *Let* $n, d$ *denote two positive integers. Let* $t > 0$ *denote a threshold parameter. Let* $\epsilon$ *denote a accuracy parameter. Given two sets of points* $A = \{a_1, \cdots, a_n\} \subset \{0,1\}^d$ *and* $B = \{b_1, \cdots, a_n\} \subset \{0,1\}^d$: *For each* $i \in [n]$, *we need to distinguish the following two cases*

- *Case 1. There exists a* $j \in [n]$ *such that* $\|a_i - b_j\|_0 < t$.

- *Case 2. For all* $j \in [n]$ *we have* $\|a_i - b_j\|_2^2 \geq (1 + \epsilon) \cdot t$.

An algorithm for $\mathsf{Gap-ANN}(n, d, t, \epsilon)$ can be called $\log(nd)$ times to binary search for the answer to ANN, so Lemma 4.3 holds as well for $\mathsf{Gap-ANN}(n, d, t, \epsilon)$.

### 4.3 Hardness Result

In the remainder of this section, we prove our lower bound for attention computation:

**Theorem 4.6** (Main Result, formal version of Theorem 1.3). *Assuming* SETH, *for every sufficiently small* $q > 0$, *there are constants* $C > 0$ *and* $C_\alpha > 0$ *and* $C_\beta > 1$ *such that Approximate Attention Computation* AAttC *(Definition 1.2) for parameters* $(n, d = C \log n, B = C_\beta \sqrt{\log n}, \epsilon_a = n^{-C_\alpha})$ *requires* $\Omega(n^{2-q})$ *time.*

*Proof.* This follows from combining Lemma 4.3 (hardness for approximation nearest neighbor search) and Lemma 4.7 (a reduction from approximate nearest neighbor search to approximate attention computation) which we prove below. $\square$

**Lemma 4.7.** *For any constant* $C_\gamma \in (0, 0.1)$: *For every* $\epsilon > 0$ *and* $C > 0$, *there exist constants* $C_a > 0$ *and* $C_b > 0$ *and such that, if* AAttC *(Definition 1.2) for parameters* $(2n, d = 2C \log n, B = C_b \sqrt{\log n}, \epsilon_a = n^{-C_a})$ *can be solved in time* $T$, *then* $\mathsf{Gap-ANN}(n, d = C \log n, t, \epsilon)$ *(Definition 4.5) can be solved in time* $O(T + n^{2-C_\gamma})$.

Due to space limitation, we defer the proof of Lemma 4.7 to Appendix B.

## 5 Conclusion

In this work, we showed that how quickly one can perform attention computation depends critically on the magnitude, $B$, of the entries of the input matrices. Our main idea was to make use of similarities between attention computation and KDE, and to show how many known techniques for KDE can also be used in this setting. Since KDE is a very well-studied problem, it would be exciting to see what other techniques can be applied to attention computation as well. One limitation of our lower bound result is, it is a conditional lower bound which is based on a well-known conjecture SETH in the area of complexity. It would be interesting to show unconditional lower bound for future work. As far as we are aware, our work does not have negative societal impacts.

**Acknowledgements** Josh Alman was partly supported by a grant from the Simons Foundation (Grant Number 825870 JA). The authors would like to thank Beidi Chen for helpful discussions related to LLMs, and Feyza Duman, Chinmay Hegde, and Piotr Indyk for helpful comments on an earlier draft. The authors would like to appreciate very constructable feedbacks for NeurIPS 2023 Reviewers. The authors would like to thanks Lichen Zhang and Ruizhe Zhang for useful and helpful suggestions about proof-reading the paper.

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

# Appendix

**Roamdap.**

We provide the missing proofs of our upper bound (algorithm) result in Section A. We provide the missing proofs of our lower bound (hardness) result in Section B.

## A Missing Proofs for Upper Bound

### A.1 Proof of Lemma 3.2

**Lemma A.1** (Restatement of Lemma 3.2). *Let $M = XY^\top \in \mathbb{R}^{n \times n}$ denote a matrix with $X, Y \in \mathbb{R}^{n \times d}$. Let $P(x)$ denote a degree-$g$ polynomial, and define $r = \binom{2(g+d)}{2g}$.*

*There is an algorithm that runs in $O(nrg)$ time and, given as input the matrix $X, Y$, constructs matrices $U_1, U_2 \in \mathbb{R}^{n \times r}$ such that $P(M) = U_1 U_2^\top$. (Here, $P(M)$ denotes the entry-wise application of $P$ to $M$.)*

*Proof.* Let $P(x)$ denote the degree-$g$ polynomial. Expand it in terms of its coefficients as

$$P(x) = \sum_{i=0}^{d} c_i \cdot x^i.$$

Consider the function $\mathsf{K} : \mathbb{R}^d \times \mathbb{R}^d \to \mathbb{R}$ defined by, for $u, v \in \mathbb{R}^d$,

$$\mathsf{K}(u, v) := P(\langle u, v \rangle).$$

$\mathsf{K}$ is a degree-$2g$ polynomial in the $2d$ entries $u_1, \cdots u_d, v_1, \cdots, v_d$ of the vectors $u, v$. Define the set $V$ of its variables,

$$V := \{u_1, \cdots, u_d, v_1, \cdots, v_d\}.$$

Let $\mathcal{F}$ denote the set of functions

$$\mathcal{F} := \left\{ f : V \to \{0, 1, 2, \cdots, 2g\} \mid \sum_{v \in V} f(v) \leq 2g \right\}.$$

We can count that

$$|\mathcal{F}| = \binom{2d + 2g}{2g}.$$

Hence, there are coefficients $c_t \in \mathbb{R}$ for each $t \in \mathcal{F}$ such that

$$\mathsf{K}(u, v) = \sum_{t \in \mathcal{F}} c_t \cdot \prod_{v \in V} v^{t(v)}.$$

Define

$$V_u := \{u_1, \cdots, u_d\}$$

and

$$V_v = V \backslash V_u.$$

We define $\phi_u : \mathbb{R}^d \to \mathbb{R}^{|\mathcal{F}|}$ by, for any $t \in \mathcal{F}$,

$$\phi_u(u_1, \cdots, u_d)_t = c_t \cdot \prod_{v_i \in V_u} u_i^{t(u_i)}.$$

Similarly, we define $\phi_v : \mathbb{R}^d \to \mathbb{R}^{|\mathcal{F}|}$ by, for any $t \in \mathcal{F}$,

$$\phi_v(v_1, \cdots, v_d)_t = \prod_{v_i \in V_v} v_i^{t(u_i)}.$$

Thus, we have

$$\mathsf{K}(u,v) = \langle \phi_u(u), \phi_v(v) \rangle.$$

For $i \in [n]$, let $X_i \in \mathbb{R}^d$ denote the $i$-th row of $X$, and let $Y_i \in \mathbb{R}^d$ denote the $i$-th row of $Y$. Our algorithm can thus construct

- the matrix $U_1 \in \mathbb{R}^{n \times |\mathcal{F}|}$ whose $i$-th row is the vector $\phi_u(x_i)$ for $i \in [n]$, and

- the matrix $U_2 \in \mathbb{R}^{n \times |\mathcal{F}|}$ whose $i$-th row is the vectors $\phi_v(y_i)$ for $i \in [n]$.

Each entry of these matrices can be constructed by multiplying together at most $g$ variables, so these $n \times r$ matrices can be constructed in time $O(nrg)$ as desired. $\qquad\square$

## A.2 Proof of Lemma 3.5

**Lemma A.2** (Restatement of Lemma 3.5). *Let $A \in \mathbb{R}^{n \times n}$ be any matrix whose entires are all positive and $\epsilon_A \in (0, 0.1)$ be any parameter. Let $\widetilde{A} \in \mathbb{R}^{n \times n}$ be any matrix such that, for all $(i,j) \in [n] \times [n]$, we have*

$$|\widetilde{A}_{i,j} - A_{i,j}| \le \epsilon_A \cdot A_{i,j}.$$

*Define the matrices $D, \widetilde{D} \in \mathbb{R}^{n \times n}$ by $D = \mathrm{diag}(A\mathbf{1}_n)$ and $\widetilde{D} = \mathrm{diag}(\widetilde{A}\mathbf{1}_n)$. Then, for all $i \in [n]$ we have*

$$|\widetilde{D}_{i,i} - D_{i,i}| \le \epsilon_A \cdot D_{i,i}.$$

*Proof.* We calculate that

$$
\begin{aligned}
|\widetilde{D}_{i,i} - D_{i,i}| = |\sum_{j=1}^{n} \widetilde{A}_{i,j} - \sum_{j=1}^{n} A_{i,j}| \\
\le \sum_{j=1}^{n} |\widetilde{A}_{i,j} - A_{i,j}| \\
\le \sum_{j=1}^{n} \epsilon_A A_{i,j} \\
= \epsilon_A \cdot D_i.
\end{aligned}
$$

where the second step follows from triangle inequality.

This completes the proof. $\qquad\square$

## A.3 Proof of Lemma 3.6

**Lemma A.3** (Restatement of Lemma 3.6). *Let $\epsilon_A, \epsilon_D \in (0, 0.1)$ and $B > 1$ be parameters, and let $V \in \mathbb{R}^{n \times d}$ denote a matrix with $\|V\|_\infty \le B$. Let $A \in \mathbb{R}^{n \times n}$ be any matrix whose entires are all positive, and let $\widetilde{A} \in \mathbb{R}^{n \times n}$ be a matrix such that, for all $(i,j) \in [n] \times [n]$ we have*

$$|\widetilde{A}_{i,j} - A_{i,j}| \le \epsilon_A \cdot A_{i,j}.$$

*Let $D, \widetilde{D} \in \mathbb{R}^{n \times n}$ be any diagonal matrices with positive entries on their diagonals, with the property that, for all $i \in [n]$, we have*

$$|\widetilde{D}_{i,i} - D_{i,i}| \le \epsilon_D \cdot D_{i,i}.$$

*Then, we have*

$$\|\widetilde{D}^{-1}\widetilde{A}V - D^{-1}AV\|_\infty \le (\epsilon_A + \epsilon_D) \cdot B.$$

*Proof.* We have

$$\|\widetilde{D}^{-1}\widetilde{A}V - D^{-1}AV\|_\infty \le \|\widetilde{D}^{-1}\widetilde{A}V - D^{-1}\widetilde{A}V\|_\infty + \|D^{-1}\widetilde{A}V - D^{-1}AV\|_\infty. \quad (1)$$

We now bound each of these two terms separately.

First, for each $(i,j) \in [n] \times [d]$,

$$
\begin{aligned}
|(\widetilde{D}^{-1}\widetilde{A}V - D^{-1}\widetilde{A}V)_{i,j}| &= |\sum_{l=1}^n (\widetilde{D}_{i,i}^{-1} - D_{i,i}^{-1}) \cdot \widetilde{A}_{i,l} \cdot V_{l,j}| \\
&\le \sum_{l=1}^n |(\widetilde{D}_{i,i}^{-1} - D_{i,i}^{-1}) \cdot \widetilde{A}_{i,l}| \cdot \|V\|_\infty \\
&= \sum_{l=1}^n |\frac{D_{i,i} - \widetilde{D}_{i,i}}{D_{i,i}\widetilde{D}_{i,i}} \widetilde{A}_{i,l}| \cdot \|V\|_\infty \\
&\le \epsilon_D \cdot \sum_{l=1}^n |\widetilde{D}_i^{-1}\widetilde{A}_{i,l}| \cdot \|V\|_\infty \\
&= \epsilon_D \cdot |\sum_{l=1}^n \widetilde{D}_i^{-1}\widetilde{A}_{i,l}| \cdot \|V\|_\infty \\
&= \epsilon_D \cdot \|V\|_\infty \\
&\le \epsilon_D \cdot B \quad (2)
\end{aligned}
$$

where the second step follows from the triangle inequality, the forth step follows from $|(D_{i,i} - \widetilde{D}_{i,i})/D_{i,i}| \le \epsilon_D$, the fifth step follows from $\widetilde{D}_i^{-1} > 0$ and $\widetilde{A}_{i,l} > 0$, and the last step follows from our assumption on $V$.

Second, for each $(i,j) \in [n] \times [d]$,

$$
\begin{aligned}
|(D^{-1}\widetilde{A}V - D^{-1}AV)_{i,j}| &= |\sum_{l=1}^n D_{i,i}^{-1}(\widetilde{A}_{i,l} - A_{i,l}) \cdot V_{l,j}| \\
&\le \sum_{l=1}^n |D_{i,i}^{-1}| \cdot |(\widetilde{A}_{i,l} - A_{i,l})| \cdot \|V\|_\infty \\
&= \sum_{l=1}^n D_{i,i}^{-1} \cdot |(\widetilde{A}_{i,l} - A_{i,l})| \cdot \|V\|_\infty \\
&\le \sum_{l=1}^n D_{i,i}^{-1} \cdot \epsilon_A A_{i,l} \cdot B \\
&= \epsilon_A \cdot B, \quad (3)
\end{aligned}
$$

where the second step follows from triangle inequality, the third step follows from $D_{i,i}^{-1} > 0$, the forth step follows from $|\widetilde{A}_{i,l} - A_{i,l}| \le \epsilon_A \cdot A_{i,l}$ and the last step follows from definition of $D_{i,i}$.

The result follows by combining Eq. (1), and two inequalities (Eq. (2) and Eq. (3)). $\qquad\square$

## B  Missing Proofs for Lower Bound

**Lemma B.1** (Restatement of Lemma 4.7)**.** *For any constant $C_\gamma \in (0, 0.1)$: For every $\epsilon > 0$ and $C > 0$, there exist constants $C_a > 0$ and $C_b > 0$ and such that, if* AAttC *(Definition 1.2) for parameters $(2n, d = 2C\log n, B = C_b\sqrt{\log n}, \epsilon_a = n^{-C_a})$ can be solved in time $T$, then* Gap$-$ANN$(n, d = C\log n, t, \epsilon)$ *(Definition 4.5) can be solved in time $O(T + n^{2-C_\gamma})$.*

*Proof.* We give an algorithm with the stated running time for Gap$-$ANN$(n, d = C\log n, t, \epsilon)$. Let $c > 0$ be a parameter we will choose later (it will be a function of $C$ and $C_\gamma$). Our algorithm will proceed to one of two cases depending on the value of $t$. If $t < c\log n$, then we will use one algorithm

which runs in time $O(n^{2-C_\gamma})$. Otherwise, if $t \geq c \log n$, we will use another algorithm which runs in time $O(T)$.

**Case 1**: $t < c \log n$.

Let $a_1, \cdots, a_n, b_1, \cdots, b_n \in \{0,1\}^d$ be the input vectors to $\mathsf{Gap-ANN}$, and let $t \in [0, d]$ denote the target distance. Recall that $d = C \log n$.

In this $t < c \log n$ case, we will simply brute-force for the answer in the following way: We first store the vectors $b_1, \cdots, b_n$ in a lookup table, then for each $i \in [n]$, we iterate over all vectors $b' \in \{0,1\}^d$ which have Hamming distance at most $t$ from $a_i$ and check whether $b'$ is in the lookup table. This determines whether there is a $b \in B$ at distance at most $t$ from $a_i$, as desired.

For each $i \in [n]$, we need to iterate over $\binom{d}{t}$ choices for the vector $b'$, so the total running time will be $O(n \cdot \binom{d}{t})$. By standard bounds on binomial coefficients, we know that

$$n \cdot \binom{d}{t} \leq n \cdot \binom{C \log n}{c \log n}$$
$$\leq n^{1+f(C,c)}$$

for some function $f : \mathbb{R}_{>0} \times \mathbb{R}_{>0} \to \mathbb{R}_{>0}$ with the property that, for any fixed $C > 0$, we have

$$\lim_{c \to 0} f(C, c) = 0.$$

We can thus pick a sufficiently small constant $c > 0$, depending only on $C_\gamma$ and $C$ such that $f(C, c) < 1 - C_\gamma$ and this entire brute-force takes $O(n^{2-C_\gamma})$ time.

**Case 2**: $t \geq c \log n$.

Let $a_1, \cdots, a_n, b_1, \cdots, b_n \in \{0,1\}^d$ denote the input of $\mathsf{Gap-ANN}(n, d, t, \epsilon)$ (Definition 4.5), and recall from Remark 4.4 that we may assume each has half its entries $0$ and half its entries $1$. We will explain how to construct an Attention matrix using this instance.

Let $C_0 \geq c$ be such that

$$t := C_0 \log n. \tag{4}$$

Let $\beta > 0$ and $\widetilde{d} \geq d$ denote parameters we will choose later (see Eq. (9) and Eq. (6), respectively). Define $\tau > 0$ by

$$\tau := \exp(\beta/2). \tag{5}$$

Intuitively, our goal in picking these parameters is that $\tau$ will be an upper bound on entries of the attention matrix, i.e., we will have:

$$\tau \geq \max_{i \in [n], j \in [n]} \exp(\beta \langle a_i, b_j \rangle / \widetilde{d}).$$

We will make use of an algorithm for the $\mathsf{AAttC}(\widetilde{n}, \widetilde{d}, B, \epsilon_a)$ problem, for the following parameters:

$$\widetilde{n} := 2n, \quad \widetilde{d} := 2d, \tag{6}$$

$$B := C_b \sqrt{\log n}, \quad \text{where} \quad C_b := \sqrt{40C/(C_0 \epsilon)}, \tag{7}$$

$$\epsilon_a := n^{-C_a}, \quad \text{where} \quad C_a := 2 + C_b^2(1 + C_0/C). \tag{8}$$

Furthermore, set

$$\beta := B^2. \tag{9}$$

We define $Q \in \mathbb{R}^{\widetilde{n} \times \widetilde{d}}$ and $K \in \mathbb{R}^{\widetilde{n} \times \widetilde{d}}$ as

$$
Q := \sqrt{\beta} \cdot \begin{bmatrix} a_1^\top & \mathbf{1}_d^\top \\ a_2^\top & \mathbf{1}_d^\top \\ \vdots & \vdots \\ a_n^\top & \mathbf{1}_d^\top \\ \mathbf{0}_d^\top & \mathbf{1}_d^\top \\ \mathbf{0}_d^\top & \mathbf{1}_d^\top \\ \vdots & \vdots \\ \mathbf{0}_d^\top & \mathbf{1}_d^\top \end{bmatrix} \quad \text{and} \quad K := \sqrt{\beta} \cdot \begin{bmatrix} b_1^\top & \mathbf{0}_d^\top \\ b_2^\top & \mathbf{0}_d^\top \\ \vdots & \vdots \\ b_n^\top & \mathbf{0}_d^\top \\ \mathbf{0}_d^\top & \mathbf{1}_d^\top \\ \mathbf{0}_d^\top & \mathbf{1}_d^\top \\ \vdots & \vdots \\ \mathbf{0}_d^\top & \mathbf{1}_d^\top \end{bmatrix}.
$$

Since each entry of $Q$ and $K$ is either $\sqrt{\beta}$ or $0$, it follows that

$$
\|Q\|_\infty \le \sqrt{\beta} = B
$$
$$
\|K\|_\infty \le \sqrt{\beta} = B
$$
$$
\|QK^\top/\widetilde{d}\|_\infty \le \frac{\beta \cdot \widetilde{d}}{\widetilde{d}} = \beta = B^2.
$$

Using the above construction of $Q \in \mathbb{R}^{\widetilde{n} \times \widetilde{d}}$ and $K \in \mathbb{R}^{\widetilde{n} \times \widetilde{d}}$, we note that

$$
A := \exp(QK^\top/\widetilde{d}) \in \mathbb{R}^{\widetilde{n} \times \widetilde{n}}
$$

is given by

$$
A = \begin{bmatrix} \exp(\beta\langle a_1, b_1\rangle/\widetilde{d}) & \exp(\beta\langle a_1, b_2\rangle/\widetilde{d}) & \cdots & \exp(\beta\langle a_1, b_n\rangle/\widetilde{d}) & \tau & \tau & \cdots & \tau \\ \exp(\beta\langle a_2, b_1\rangle/\widetilde{d}) & \exp(\beta\langle a_2, b_2\rangle/\widetilde{d}) & \cdots & \exp(\beta\langle a_2, b_n\rangle/\widetilde{d}) & \tau & \tau & \cdots & \tau \\ \vdots & \vdots & \ddots & \vdots & \vdots & \vdots & \ddots & \vdots \\ \exp(\beta\langle a_n, b_1\rangle/\widetilde{d}) & \exp(\beta\langle a_n, b_2\rangle/\widetilde{d}) & \cdots & \exp(\beta\langle a_n, b_n\rangle/\widetilde{d}) & \tau & \tau & \cdots & \tau \\ \exp(0) & \exp(0) & \cdots & \exp(0) & \tau & \tau & \cdots & \tau \\ \exp(0) & \exp(0) & \cdots & \exp(0) & \tau & \tau & \cdots & \tau \\ \vdots & \vdots & \ddots & \vdots & \vdots & \vdots & \ddots & \vdots \\ \exp(0) & \exp(0) & \cdots & \exp(0) & \tau & \tau & \cdots & \tau \end{bmatrix}.
$$

(Note that we do not explicitly compute all the entries of $A$ in our algorithm; we will make use of it only through calling our algorithm for the Attention problem.)

For each $(i, j) \in [n] \times [n]$, we know that

$$
\begin{aligned}
A_{i,j} &= \exp(\beta\langle a_i, b_j\rangle/\widetilde{d}) \\
&\le \exp(\beta\|a_i\|_\infty \cdot \|b_j\|_\infty \cdot d/\widetilde{d}) \\
&\le \exp(\beta) \\
&= \tau
\end{aligned} \tag{10}
$$

where the first step follows from definition of $A \in \mathbb{R}^{\widetilde{n} \times \widetilde{n}}$, the second step follows from $\langle a_i, b_j\rangle \le \|a_i\|_\infty \cdot \|b_j\|_\infty d$, the third step follows from $d < \widetilde{d}$ (see Eq. (6)), and the last step follows from definition of $\tau$ (see Eq. (5)).

On the other hand, we know that that for each $(i, j) \in [n] \times [n]$,

$$
A_{i,j} \ge 0 \tag{11}
$$

since it is an exponential of an entry of $QK^\top/\widetilde{d}$.

Using Eq. (10) and Eq. (11), combined with our expression for $A$, it thus follows that

$$
n\tau \le (A\mathbf{1}_{\widetilde{n}})_i \le 2n\tau, \quad \forall i \in [\widetilde{n}].
$$

Since $D_{i,i} = (A\mathbf{1}_{\widetilde{n}})_i$, thus we know that
$$n\tau \leq D_{i,i} \leq 2n\tau, \quad \forall i \in [\widetilde{n}].$$

Choose the vector $v \in \mathbb{R}^{\widetilde{n}}$ defined as
$$v = \begin{bmatrix} \mathbf{1}_n \\ \mathbf{0}_n \end{bmatrix}.$$

We define $\widetilde{t}$ as
$$\widetilde{t} := \frac{1}{3}\exp(0.25\beta(1 - t/d))/(2n\tau). \tag{12}$$

We can show that $\widetilde{t} \geq \epsilon_a$ as follows:
$$\begin{aligned}
\widetilde{t} &= \frac{1}{6n}\exp(0.25\beta(1 - t/d) - \beta) \\
&= \frac{1}{6n}\exp(-0.75\beta - 0.25\beta t/d) \\
&= \frac{1}{6n}\exp(-0.75\beta - 0.25\beta C_0/C) \\
&= \frac{1}{6}\exp(-0.75\beta - 0.25\beta C_0/C - \log n) \\
&= \frac{1}{6}\exp(-0.75C_b^2 \log n - 0.25C_b^2(C_0/C)\log n - \log n) \\
&\geq n^{-C_a} \\
&= \epsilon_a,
\end{aligned}$$
where the second step follows from simple algebra, the third step follows from $t = C_0 \log n$ (Eq. (4)) and $d = C \log n$ (assumption in Lemma statement), the second step follows from choice of $\beta$ (Eq. (7)), and the sixth step follows from choice of $C_a$ (Eq. (8)), and the last step follows from Eq. (8).

Since $\widetilde{t} \geq \epsilon_a$, if we run an algorithm for Approximation Attention Computation (Definition 1.2) $\mathsf{AAttC}(\widetilde{n}, \widetilde{d}, B, \epsilon_a)$, where we pick $V$ to be a matrix with one row $v$ and the rest $0$, we can output a vector $u \in \mathbb{R}^{\widetilde{n}}$ such that, for all $i \in [\widetilde{n}]$,
$$|u_i - (D^{-1}Av)_i| < \widetilde{t}.$$

Note that using Remark 4.4, we have
$$\begin{aligned}
\|a_i\|_2^2/d &= 0.5, \quad \forall i \in [n], \\
\|b_j\|_2^2/d &= 0.5, \quad \forall j \in [n].
\end{aligned}$$

Therefore, for any $(i, j) \in [n] \times [n]$,
$$\begin{aligned}
\frac{1}{d}\langle a_i, b_j \rangle &= \frac{1}{2d}(\|a_i\|_2^2 + \|b_j\|_2^2 - \|a_i - b_j\|_2^2) \\
&= \frac{1}{2d}(0.5d + 0.5d - \|a_i - b_j\|_2^2) \\
&= 0.5 - 0.5\|a_i - b_j\|_2^2/d,
\end{aligned}$$
where the second step follows from $\|a_i\|_2^2 = \|b_j\|_2^2 = d/2$, and the last step follows from simple algebra.

Recall that our goal is to determine, for each $i \in [n]$, whether there is a $j \in [n]$ such that $\|a_i - b_j\|_2^2 \leq t$, or whether $\|a_i - b_j\|_2^2 \geq (1 + \epsilon_a)t$ for all $j \in [n]$. We will show next that we can distinguish these two cases by seeing whether $u_i$ is greater than or less than the value $\widetilde{t}_0 := 2\widetilde{t}$.

**Case 2a.**

If there exists an $(i,j) \in [n] \times [n]$ such that $\|a_i - b_j\|_2^2 \leq t$, then

$$\beta \langle a_i, b_j \rangle / \widetilde{d} = 0.5 \cdot \beta \langle a_i, b_j \rangle / d$$
$$\geq 0.25 \cdot \beta(1 - t/d),$$

where the first step follows from $2d = \widetilde{d}$ (see Eq. (6)).

This means that

$$u_i \geq \exp(0.25\beta(1 - t/d))/(2n\tau) - \widetilde{t}$$
$$= 3\widetilde{t} - \widetilde{t}$$
$$= 2\widetilde{t}$$
$$= \widetilde{t}_0,$$

where the second step follows from the definition of $\widetilde{t}$ (see Eq. (12)), and the last step follows from the definition of $\widetilde{t}_0$.

**Case 2b.**

If for all $(i,j) \in [n] \times [n]$, we have $\|a_i - b_j\|_2^2 > t(1 + \epsilon)$, this implies

$$\beta \langle a_i, b_j \rangle / d \leq 0.25\beta \cdot (1 - t(1 + \epsilon)/d).$$

Then, for all $i \in [n]$,

$$u_i < (n \cdot \exp(0.25\beta(1 - (1 + \epsilon)t/d)))/(n\tau) + \widetilde{t}$$
$$= \exp(0.25\beta(1 - t/d))/(2n\tau) \cdot (n/\exp(0.25\beta\epsilon t/d)) + \widetilde{t}$$
$$= 3\widetilde{t} \cdot (n/\exp(0.25\beta\epsilon t/d)) + \widetilde{t}$$
$$\leq 3\widetilde{t} \cdot \frac{1}{4} + \widetilde{t}$$
$$= 2\widetilde{t}$$
$$= \widetilde{t}_0,$$

where the third step follows from definition of $\widetilde{t}$ (see Eq. (12)), the forth step follows from the calculation in Eq. (13) below, and the last step follows from $\widetilde{t} = \widetilde{t}_0/2$.

Finally, b our choice of $\beta$ and $t$, we can see that

$$\exp(0.25\beta\epsilon t/d) = \exp((0.25\beta\epsilon C_0 \log n)/d)$$
$$= \exp(0.25\beta\epsilon C_0/C)$$
$$= \exp(10 \log n)$$
$$> 4n, \tag{13}$$

where the first step follows $t = C_0 \log n$ (Eq. (4)), the second step follows from $d = C \log n$, and the third step follows from $\beta = B^2$ (Eq. (9)) and the choice of $B$ (Eq. (7)). $\square$

**Recent Followup Work** During the preparation of camera ready, we are aware of a number of new related work [KMZ23, HJK$^+$23, AS23, DLS23, GSWY23, DSZ23, GSY23] to this paper. This work studies softmax attention, and [KMZ23] shows to speed up the polynomial based attention unit via sketching techniques. [HJK$^+$23] introduces hyperattention, which is able to handle long-context attention in near-linear time. This work is mainly focusing classical attention computation, [AS23] consider a more general tensor version computation of attention scheme. This work studies the inference computation, the recent work [DLS23, GSWY23] also considers how to speedup the training process which involves the computation of gradient and hessian. Inspired by hardness result of this work, [DSZ23] shows that there is a binary classification tasks (involves two datasets) that softmax attention can distinguish but linear attention cannot. [GSY23] provides a result for computing the attention matrix differentially privately. [GSYZ23] introduce a quantum algorithm for computing attention matrix.

