# OpenReview forum: "Fast Attention Requires Bounded Entries"
_NeurIPS.cc/2023/Conference — NeurIPS 2023 poster_

### Official Review · Reviewer_LkJK · 2023-07-02

**Soundness:** 2 fair
**Presentation:** 3 good
**Contribution:** 2 fair
**Rating:** 6
**Confidence:** 3

**Summary:**

This paper shows that whether or not there exists a fast algorithm for approximately computing the attention matrix ($A \in \mathbb{R}^{n \times d}$, where $d = O(\log n)$ depends on the bound of the entry value $B$

1. Particularly, the authors show that when $B = o(\sqrt{\log n)}$, their proposed algorithm can compute the attention matrix within $1/\text{poly}(n)$ error in time $n^{1+o(1)}$.
2. Assuming the Strong Exponential Time Hypothesis (SETH) and the fine-grained complexity conjecture is valid, they show that there is no $O(N^{2-q})$ time algorithm that guarantees $1/\text{poly}(n)$ approximation error.

**Strengths:**

+ The results are interesting and correct
+ An improved hardness result is a good contribution
+ The observation that the existence of a sublinear algorithm depends on the entry value is interesting.

**Weaknesses:**

The proposed algorithm is a bit incremental, both in terms of the proposed method, and the techniques to establish these results.

Minor typo:
p5: In the proof of Corollary 2.2, it seems like $\max$ operation is missing in the formula $g = \Theta(...)$.

**Questions:**

+ Although the paper provides an explicit way of the matrix construction $U_1$ and $U_2$, the reviewer is wondering how complex is the matrix construction in practice.

**Limitations:**

+ It would be nice if there were simulation results that demonstrate the practicality for some benchmark datasets on some target tasks.

---

> ### Author Rebuttal · Authors · 2023-08-07
>
> We thank the reviewer for the thoughtful review.
>
> Minor typo p5: Yes, it is a typo and the max operator is missing. Thank you for pointing it out. We will fix it in the updated version.
>
> Question: The construction of U_1 and U_2 in Lemma 3.2 is actually very simple and practical. At a high level: each column of U_1 corresponds to a monomial of the polynomial, each row corresponds to an input vector, and the corresponding entry of U_1 is equal to that monomial evaluated at that input. (U_2 is similar.) Thus, U_1 and U_2 can be constructed in essentially the same amount of time as it takes to just write down their entries. See Appendix A.1 in our supplementary materials for the full details.

---

> > ### Comment · Reviewer_LkJK · 2023-08-17
> >
> > I would like to thank the authors for answering the question. I remain to be on the positive side for this submission.

---

### Official Review · Reviewer_ZPux · 2023-07-03

**Soundness:** 3 good
**Presentation:** 3 good
**Contribution:** 4 excellent
**Rating:** 7
**Confidence:** 4

**Summary:**

The paper is a purely theoretical paper discussing improving the computational time required to solve the attention computation problem which is known to be quadratic in time. The idea hinges upon the use of the boundness of the entries of the matrices involved in such a problem.
The authors discuss mainly two cases, where the entries are bounded by a polylogarithmic term in the number of tokens which then the attention computation problem can be solved in time almost linear in the number of tokens. Otherwise, the authors prove via the "Strong Exponential Time Hypothesis" from fine-grained complexity theory that the running time is no better than quadratic in the number of tokens.

**Strengths:**

1) The paper is well-written and easy to follow.
2) The paper proves the existence of almost linear time algorithms for the attention computation problem, as it depends on the boundness of the entries of the matrices.
3) The paper also shows that in case the entires of the involved matrices in the attention computation problem are "not bounded", then the quadratic time associated with the attention computation problem can not be improved.

**Weaknesses:**

The only weakness is the lack of practical evaluation. Although the paper sets a theoretical justification for, e.g., the use of quantization or low-degree polynomial approximation of the attention computation problem, it would be great to observe the quality of the proposed algorithm on the quality of transformer-like models which practices the attention computation problem.

**Questions:**

1) On page 14 of the appendix, did you mean to put n instead of D at the top of the summation symbol (see equation 2)?
2) Also on the same page, did you mean to write $\tilde{D}_{i, i}$ instead of $\tilde{D}_i$?
3) Shouldnt be the $B$ in the denominator of the equation stated at Lemma 3.4 be $B^2$?

**Limitations:**

The authors adequately addressed the limitations of their work.

---

> ### Author Rebuttal · Authors · 2023-08-07
>
> We thank the reviewer for the thoughtful review. We really appreciate the reviewer for pointing out these typos, we will fix them in the next version. (We remark that none of the typos impact the correctness of the paper.)

---

### Official Review · Reviewer_Ze7L · 2023-07-07

**Soundness:** 2 fair
**Presentation:** 3 good
**Contribution:** 2 fair
**Rating:** 3
**Confidence:** 4

**Summary:**

In this work, the author studied the problem of how to efficiently compute or approximate the attention matrix. The major contribution can be summarized as:

1. they found for the case when $d = O(\log n)$ and entries of the input matrices are uniformly bounded by $o(\sqrt{\log n})$, then an algorithm that approximates the attention matrix can be constructed with sub-quadratic complexity, the key idea of the approximation is based on a polynomial approximation of the $e^x$ that was introduced by another work [AA22].

2. when $d = O(\log n)$, and the entries of the input matrices are of the order $\Theta(\sqrt{\log n})$, then under the Strong Exponential Time Hypothesis, it is impossible to approximate the attention matrix with small error and sub-quadratic complexity.




**Strengths:**

1. I think in this work, the author did a good job explain the basic intuitions and ideas behind their approach. Especially, breaking down the proof steps into smaller parts improved the readability of this work.

2. On the other hand, by elaborating the related works and contributions, the reviewer can have a better understanding of the contribution of this work.

**Weaknesses:**

There are several weakness of this work:

1. The major contribution of this work is not emphasized enough or minor. For example, the technical tools used in this work are developed by previous works to tackle similar problems. As a theory paper, I think the author at least needs to address what technical contribution is made in this work. To me, it seems that the major results are straightforward applications of some closely related recent works.

2. The lack of simulation results is another weakness. As the author is trying to address some practical/computational aspects of the problem, experimental results are essential to validate the theory and claims.

3. Some of the claims in the theorem are ambiguous. For instance, in theorem 4.6, there are many constants $C_a, C_{\beta}, C, q$ used. What are those constants? Numerical constants? What are the relationships between them? Statements like this is not rigorous at all for theory papers.

**Questions:**

Please see some questions I mentioned in the weakness section.

Aside from that, can the author explain the claim "This gives a theoretical explanation for the phenomenon observed in practice that attention computation is much more efficient when the input matrices have smaller entries". Why the results in this work explain this in practice?

**Limitations:**

Please see the weakness section.

---

> ### Author Rebuttal · Authors · 2023-08-07
>
> We thank the reviewer for the thoughtful review.
>
> 1. We discuss our technical contributions and how they compare with prior work in sections 1.1 and 1.2 of our paper. Perhaps the biggest technical insight is that a number of tools from the literature on KDE, which were not previously known to be applicable to Attention, can in fact be applied in this setting. (Indeed, prior work like [ZHDK23] and [KWH23] knew about the KDE connection but was not able to prove such strong results.) We also believe that the details of using these techniques in this setting are non-trivial. For instance, we initially thought that proving a lower bound in this way would be impossible because of the normalizing factor D^{-1} which appears in Attention computation (Definition 1.1 in our paper): previous lower bound reductions for KDE requiring finding large entries of the output, but the normalizing factor ensures that all the entries are roughly equal to each other.
>
> 2 and Questions: please see the general remark to reviewers above.
>
> 3: All our theorems and lemmas are fully formal, precise and rigorous mathematical statements. For instance, for the mentioned Theorem 4.6, we give a fully quantified logical statement (for all q, there exist constants C, C_alpha, C_beta, such that …) and all the variables in the statement are captured by the quantifications.
>
> The dependance of the constants C, C_alpha, C_beta on q is an interesting question which touches on a major open challenge in fine-grained complexity theory. The statement of the Strong Exponential Time Hypothesis [IP01] says that for every eps > 0, there exists a k such that k-SAT cannot be solved in time 2^{(1-eps)n} (see Hypothesis 4.1 in our paper). The relationship between eps and k is a major open problem; the current best SAT algorithms achieve eps = 1/O(log k), although it is generally believed that there should be better SAT algorithms that improve on this dependence. This unknown propagates to yield unclear relationships between constants in most fine-grained complexity results. For instance, it also appears in hardness for approximate nearest neighbors [Rub18] (see Lemma 4.3 in our paper), and thus in turn appears in our results which build off of [IP01] and [Rub18].
>
> That said, some relationships between the constants do appear in our proofs; see, e.g., line 518 in our supplementary material. We don’t think the details of these constants are particularly salient, and we chose not to emphasize them in our theorem statement to make it more readable.
>
> (The references [IP01] and [Rub18] are cited in our submission, but we copy them here below for convenience.)
> [IP01] Russell Impagliazzo and Ramamohan Paturi. On the complexity of k-sat. Journal of Computer and System Sciences, 62(2):367–375, 2001.
>
> [Rub18] Aviad Rubinstein. Hardness of approximate nearest neighbor search. In Proceedings of the 50th annual ACM SIGACT symposium on theory of computing (STOC), pages 1260–1268, 2018.

---

### Official Review · Reviewer_ioBT · 2023-07-26

**Soundness:** 4 excellent
**Presentation:** 4 excellent
**Contribution:** 3 good
**Rating:** 6
**Confidence:** 2

**Summary:**

The attention mechanism, whose runtime complexity is approximately quadratic in the sequence length $n$, is the main bottleneck to LLM inference. It is thus quite reasonable to study whether this can be *approximated* with sufficient accuracy in subquadratic time. In particular, for this paper the authors define and study the problem of approximate attention computation (AAttC), wherein one seeks a result that is accurate within a pre-specified entrywise error tolerance $\epsilon_a$.

Under what I think are reasonable assumptions, the authors find that whether AAttC can run in subquadratic time hinges on the bound $B$ on the magnitudes of the entries of $Q$/$K$/$V$. This is slightly surprising and, in my opinion, a neat story for the paper. This conclusion consists in two results:

1. Theorem 3.8: an algorithm that computes AAttC in near-linear time provided that $B = o(\sqrt{\log n})$. The high-level idea is
   - Approximate the $\exp$ in $\exp(QK^T/d)$ with a polynomial $P$ (Lemma 2.1 from [AA22]).
   - Compute a low-rank decomposition $U_1 U_2^T = P(QK^T/d)$ directly in linear time based on knowledge of $P$ and the entries of $Q$ and $K$ (Lemma 3.2).

    These lemmas are the main workhorses of Theorem 3.8, but effort is also made stitching everything together. The crucial thread is that $B^2$ determines the degree of $P$, which in turn determines the rank of the decomposition $U_1 U_2^T$. $B$ thus needs to grow slowly in $n$ to make the algorithm subquadratic.
2. Theorem 4.6, which shows that if $B = \Theta(\log n)$, there is no subquadratic-time algorithm for AAttC (assuming the strong exponential time hypothesis, SETH). This result is based on a reduction of approximate nearest neighbor search to AAttC.

**Strengths:**

- Although past work has produced some closely related results (ZHDK23, KWH23), the authors' focus on $B$, the entrywise bound, is original and insightful. Furthermore, the proof techniques in this paper bring up interesting connections to the KDE literature which may be interesting for the community.
- They provide a near-linear time attention algorithm based on $B = o(\sqrt{\log n})$, and show that the rate at which $B$ scales cannot be improved upon. In some sense their work is a complete description of the problem, conditional on their (reasonable) assumptions on how other quantities scale with $n$ ($d = O(\log n), \epsilon_a = 1/\text{poly}(n)$). This is quite satisfying.
- For a theoretical paper, this is well-written and easy to understand. The authors do a great job of motivating, roadmapping, and explaining their work.
- This work offers a key insight about an important problem.


**Weaknesses:**

This is a pair of complexity results. It is not clear to me how the insights provided here can be operationalized to improve AAttC in a practical setting. (The authors do highlight practical examples in ZBIW19 and KVPF20, but the connections between those works and the theory developed here do not seem particularly deep).

Typos
- L117: Hedge -> Hegde
- L133: algoriths -> algorithms
- L134: I believe should read "To design our algorithm for *Theorem 1.4*" not Theorem 1.3.
- L242: entires -> entries
- L232, L257, Algorithm 1 L4: I believe $B$ in $\frac{\log(1/\epsilon)}{\log(\log(1/\epsilon)/B)}$ was meant to be $B^2$
- Appendix L434: roamdap -> roadmap
- Appendix L453: $v_i$ -> $u_i$ under the product sign


**Questions:**

I am curious to hear how you think your results might lead to practical speedups in attention computation for future work.

**Limitations:**

The authors do highlight their main limitation: Theorem 4.6's reliance on the strong exponential time hypothesis (SETH). As far as I can tell, no important limitations were omitted.

---

> ### Author Rebuttal · Authors · 2023-08-07
>
> We thank the reviewer for the thoughtful review. We really appreciate the reviewer for pointing out these typos, we will fix them in the next version. (We remark that none of the typos impact the correctness of the paper.)

---

> > ### Comment · Reviewer_ioBT · 2023-08-14
> > **Thanks for the responses**
> >
> > I have read the authors' rebuttals and the other reviews. I maintain the view that this work should be accepted.

---

### Author Rebuttal · Authors · 2023-08-07

We want to thank the reviewers for their thoughtful and detailed reviews.

We want to briefly expand on the connection between our result and practical papers [ZBIW19] and [KVPF20] which multiple reviewers asked about. At a high level, these two papers showed that techniques for enforcing and manipulating bounded entries can quickly perform attention computations in practice. A main motivating question for our paper is whether there is a theoretical basis for this improvement. We show that there is a fast algorithm with provable guarantees in this setting, but also that bounded entries are theoretically _required_ for a fast algorithm. In other words, we show that bounded entries are not just a heuristic used by the prior work, but that they are both necessary and sufficient for fast algorithms with provable guarantees.

---

### Decision · Program_Chairs · 2023-09-21

**Decision:**

Accept (poster)

**Comment:**

This paper presents an analysis for the phenomenon that attention computation is much more efficient when the input matrices have smaller entries. While there is a discrepancy in the reviews, we believe that this work makes some solid technical contributions (on the theoretical side) to an important practical question. We hope the reviews and feedbacks during the discussions could be helpful for the authors to prepare the final version of this paper.